# Perceptions of Electric Scooters Prior to Legalisation: A Case Study of Dublin, Ireland, the 'Final Frontier' of Adopted E-Scooter Use in Europe

Páraic Carroll 

School of Civil Engineering, University College Dublin, D04 V1W8 Dublin, Ireland; paraic.carroll@ucd.ie; Tel.: +353-17-163-215

**Abstract:** This paper presents the results of a study conducted to examine pre-COVID-19 travel patterns, mode choice, and perceptions and attitudes specifically in relation to micromobility devices, namely e-scooters, in Dublin, Ireland. Given the novelty of this mode of transport and the notable current absence of e-scooter companies operating in Ireland (due to the legal status of such devices in Ireland at the time of writing), user data on electric scooters are lacking in the context of Ireland, which presents challenges for government and local authorities to develop appropriate regulations to legislate for their use. In this study, a survey was created that targeted individuals that live and work in the county of Dublin. The survey was used to examine the sociodemographic and travel characteristics influencing mode choice in Dublin; to generate an understanding of the existing and potential future demand for electric scooters; and to determine the perception of e-scooters amongst Dubliners. The main findings generated from the analysis of the survey results were the following: time followed by convenience were two of the main factors that influence mode choice, females were found to be willing to pay more than males for a shared e-scooter service, respondents with a daily trip cost of €1–5 were found to be willing to pay €4 or more for a shared e-scooter service, and 31% of respondents with a travel cost of €1 or less would be prepared to pay €2–3 for the scheme. These findings suggest that people would be willing to increase their daily travel costs in order to use the shared e-scooter service. Younger individuals on high incomes that were not in possession of a private car or a driver licence were found to be more likely to choose an e-scooter, and shorter trips (shorter distance and time) were associated with e-scooter mode choice. The results also determined that while the people in this sample of those living and working in Dublin recognise the benefits that e-scooters present to users and generally hold positive attitudes towards them, they are also wary of how they will be legislated for from a regulatory point of view in relation to speed limits, age limits, and legal riding zones to reduce the incidences of dangerous riding and collisions on roads and footpaths.

**Keywords:** micromobility; e-scooters; e-bikes; active modes

## 1. Introduction

Micromobility and, more specifically, electric scooters have become a contentious topic of debate for many policy makers and local government officials in many cities around the world as they grapple with devising regulations and by-laws and weigh up the advantages and disadvantages of their use in urban settings. Most cities and countries who have legalised for the use of e-scooters have imposed a 25 km/h or 15 mile/h speed limit, prohibit their use on pavements and, depending on the city, have different parking restrictions, i.e., docked, virtual docked or free-floating approaches. In the United Kingdom, e-scooters can currently only be used on public roads if they are part of the various government approved trials that are taking place across the country, as they are considered 'powered transporters', and personal (private) use is only permitted on private property. In other words, it is illegal

to ride an electric scooter in public unless it is a shared e-scooter that is included in the government-led trials. In the Republic of Ireland, the situation is quite similar to that in the UK, where e-scooters are termed as 'powered personal transporters' and their use on public roads are illegal; however, there are currently no public e-scooter trials taking place or approved on Irish roads, meaning that it is only possible to ride an e-scooter on private lands. However, notwithstanding the illegal status of e-scooters in Ireland, they are legally sold in shops and are a popular mode of transport used by many. Thus, it is worth noting that the popularity of this mode is such that those riding such devices are willing to risk confiscation or a fine from the police in order to derive the benefits that e-scooters provide to users.

This delineated situation, which has been the case for approximately the past 4 years in the Republic of Ireland, has been one of the main motivations to conduct this study, which has the objective of collecting data on pre-COVID-19 travel patterns, mode choice, and perceptions and attitudes towards micromobility, specifically e-scooter use, in Dublin, Ireland. Data on electric scooters and micromobility is lacking in the context of Ireland, which presents challenges for central government and local authorities to develop appropriate legislation and amend road traffic regulations. Ireland is also one of the last countries in Europe to legislate for the use of e-scooters on public roads, which has generated pent-up competition and interest amongst micromobility operators who are seeking to secure an operating licence there. While it is legal to purchase and sell such devices in Ireland, the current lack of legislation acts a barrier to the wider adoption of this mode amongst many people, which is rapidly growing in popularity. Some of the primary issues or concerns related to e-scooter use that have been dividing opinion are traffic accidents and injuries involving e-scooter riders [1], accessibility of e-scooters to underage or young children/teenagers and, on occasion, dangerously riding them on roads, riding e-scooters on footpaths/sidewalks, in-built speed limiter tampering in scooters and e-bikes, and cases of reckless parking of dockless e-scooters in different European cities. Such issues are high on the agenda in terms of regulation proposals in Ireland, and many micromobility operators are closely examining and implementing solutions to addressing such issues [2]. However, in October 2021, draft legislation (entitled The Road Traffic and Roads Bill 2021) was released, which, if passed by central government, will legalise the use of such devices termed 'powered personal transporters (PPTs)' on public roads in the Republic of Ireland. This also coincides with a strong demand for private e-scooters sales as mentioned above, which are frequently seen being used on the streets of the main regional cities of the country, such as Dublin, Cork, Galway, Limerick, and Waterford, despite their current illegal status. Ireland's first e-scooter trial (on private lands) was launched in October 2019 on the Dublin City University (DCU) campus, where shared micromobility operator Blue Duck began trialling the use of computer vision-enabled e-scooters on the campus, in partnership with Luna (an Irish tech company providing Edge AI and computer vision solutions) [3].

Some notable recent studies have examined e-scooter use and the service characteristics of shared micromobility operations. For example, when examining attitudinal factors of continued use of e-scooters in Chicago, Javadinar et al. [4] found via a Partial Least Square Structural Equation Model (PLS-LEM) that perceived usefulness and reliability (i.e., a guaranteed availability) of e-scooters were the main influencing factors affecting users' choice to opt for shared e-scooters. The UK Department for Transport commissioned a study into the perceptions and preferences of e-scooters in the UK and found that 7% of the sample of respondents (total of 4000) had used an e-scooter; that males in the 16–24 age group were the most likely users; and that the majority of user reasons for riding an e-scooter was for fun, followed by commuting or accessing local amenities [5]. Furthermore, 60% of respondents supported the legalisation of riding e-scooters in cycle lanes in the UK, with 14% stating that they should not be permitted for use in any public areas. The findings from this study in addition to other similar research, such as by Christoforou, et al. [6] in Paris, France, also showed that many e-scooter users shifted from walking and public transport

modes rather than private cars. This is a subject that will be examined further in this paper by investigating the results of the survey, which include several attitudinal questions.

Ease of use and enjoyment were found be important factors in future e-scooter use. Oeschger et al. [7] found that there is a lack of empirical work which examines the integration of micromobility and public transport and the economic, societal, and environmental benefits that such integration can provide, particularly in relation to how micromobility can cater for longer access and egress trips. Access and egress trips or first and last mile segments were a particular focus in the study presented in this paper.

In a study which applied a nested choice model to data from four shared micromobility companies in Zurich, Switzerland, Reck et al. [8] determined that docked services were more popular during peak hours of the day, while the dockless scheme had a higher uptake during off-peak hours, which is an interesting finding that may prove useful for micromobility companies in fleet management and planning models. Similarly, e-bikes were found to be the preferred micromobility mode for commuting purposes, while electric scooters were more popular for other trips.

A study conducted by a Dublin hospital [9] which examined medical records of 22 patients with injuries associated with e-scooter use found that in the case of Dublin, e-scooter trauma results in a 'high rate of orthopaedic injuries' often requiring surgical attention, which was compounded by 60% of the sample not wearing a helmet and 73% requiring physiotherapy. This is a growing area of interest in the research community as evidenced recently [1], which was found to be the main negative aspect of e-scooters reported by the sample in this study, as presented later in this paper.

However, there is also evidence from some transport agencies in cities in the United States that shows that 38% of Portland e-scooter riders and 48% of visitors to the city used an e-scooter rather than private car or on-demand vehicles (Uber/taxi) [10], while the San Francisco Municipal Transportation Agency [11] reported that 42% of e-scooter users would have travelled by private car if an e-scooter was not an available mode choice. Thus, there are conflicting reports in different jurisdictions on the topic of whether e-scooters encourage a mode shift from private cars or from active modes and public transport, which suggests that this is a much more complex discussion rooted in the specific characteristics of public transport and active mode networks, private vs. shared e-scooter use, levels of traffic congestion, and road space allocation amongst, other attributes. It is therefore not straight forward to assume mode shift vs. another unless local research and data are available to support such a notion. This question will also be explored in the context of Dublin, as presented later in this paper.

The study presented in this paper seeks to address the following research objectives: examine the public perception and attitude towards e-scooters ahead of the passing of key legislation in Ireland and the latent demand for e-scooters and the socio-demographic and trip characteristic variables that are linked to e-scooter mode choice. The data collection phase of the research conducted consisted of a survey/questionnaire, which was devised to determine the existing awareness and potential demand of electric scooters in Dublin in advance (at the time of writing) of the passing of the Road Traffic Bill, which, if enacted, will allow for the introduction of appropriate regulations and legislation for e-scooter use in Ireland. In addition to this, this study examines the sociodemographic and travel characteristics which influence the mode choice of the sample, to determine the public perception of e-scooters in Dublin, which may provide an insight into the public consciousness and appetite (or not) for micromobility, and potentially inform policy/regulatory decision making in this area.

This paper is ordered in four sections. The first section has introduced the context for the research examined in the paper and provides a brief review of relevant studies. Section 2 outlines the structure of the survey conducted and methodology adopted to analyse the survey results, which are presented in Section 3, before being discussed further in the discussion and conclusion in Section 4.

## 2. Methods

The study area for this research was County Dublin, which was selected for this study due to the fact that there is a greater assortment of alternate and sustainable transport modes available in this region in comparison to the rest of Ireland, which presents more viable opportunities for micromobility to encourage longer-distance multi-modal trips when linked with public transport services.

Figure 1 below displays the study area and the electoral districts where the respondents reside, illustrated in graduated colours, with the areas displayed in green and yellow having more respondents than the areas displayed in red and orange. This map shows that there is strong representation across the majority of the electoral divisions in Dublin city and county, with a relative balance between the north and south side of the county and without an evident focus on the inner city.

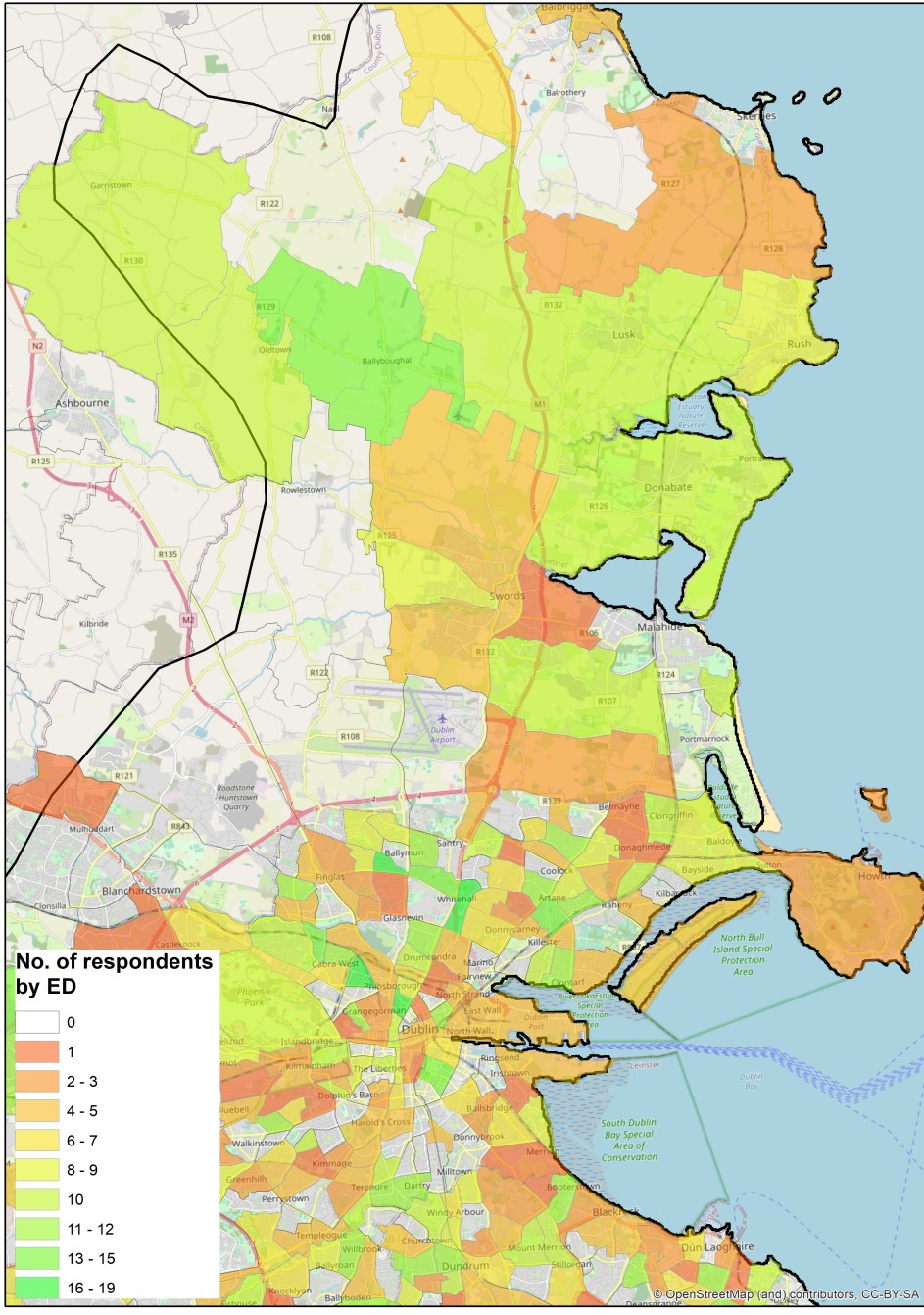

**Figure 1.** Living location of respondents on Electoral District level.

The survey was conducted online in early February 2021, using the survey company 'Bounce Insights' [12], and was distributed randomly to a representative sample of the population residing and working or studying in Dublin City and County (survey area illustrated in Figure 1). Bounce Insights provided the tool to develop the online survey instrument and disseminate the survey to the respondents. To encourage the recruitment of potential respondents, Bounce offers an innovative reward-based scheme which incentivises people to respond to various surveys by rewarding them in the form of points/coupons that can be accumulated and then 'cashed in' at online retailers or sites.

The survey administered was organised into 5 distinct parts. The first section, consisting of introductory or screener questions, initially screened the respondents before undertaking the survey. This was conducted in order to firstly ensure that that the respondents were briefed on the topic of the survey and that the answers provided should be in relation to pre-COVID-19 conditions and not conditions during prevailing lockdown restrictions, and secondly, to ensure that those who responded to the survey were residing and working or studying in Dublin City and/or County. The second section of the survey presented a series of questions which mirrored those used in the Census of Ireland questionnaire, as a means of further measuring the representativeness of the sample collected. These questions were used to elicit the trip characteristics of the sample, such as marital status, employment status, highest level of education, number of dependents/children, possession of a driver licence, etc. The responses to such questions were then used to develop an understanding of the demographic profile of those interested or uninterested in shared electric scooters, and also to determine the influences on mode choice in Dublin. The third part of the questionnaire surveyed the respondents' current transport mode choice, journey length (time and distance), daily cost of travel, car ownerships levels, and access and egress distances to and from public transport stops/stations for commute trips to work or education. Such responses were utilised to examine existing travel practices and potential for mode shift in Dublin with the presence of micromobility. The fourth section then asked the sample to state their experience with e-scooters, i.e., whether they have used one before or have they witnessed/seen others riding them. This was followed by attitudinal questions, which produced insights in relation to perceptions and/or attitudes that Dubliners have of e-scooters, such as what they believe to be the benefits or drawbacks of electric scooters. These questions were designed to encourage the respondents to reflect on whether they would have a desire to use a shared scooter service in Dublin. The final section of the survey then provided an opportunity for the respondents to express their opinion on the electric scooter debate and to provide any feedback that they would like e-scooter operators to consider.

## 3. Results

Sample Characteristics

According to the population of County Dublin (1.347 million, Census [13]), with a 95% confidence level and a 5% margin of error, 400 respondents were found to be a representative sample size for this study. In total, as shown in Table 1, 431 responses to the survey were recorded, of which 44% were female and 55% male, 38% were in the 18–24 age cohort, 37% were in the 25–39 bracket, and 25% of the sample were over 40 years of age. The mean age of the sample is largely in line with the mean age of the population of Ireland (i.e., 36.5 years), according to the 2016 Census, which adds to the representativeness of the sample recorded.

The average annual income range of each respondent was similarly recorded, which revealed that 44% of the sample earned €25,000 or less, followed by 37% earning between €25,000 and €50,000 per year. Only 5% of the sample earned over €75,000 per annum. This is largely representative of the median annual earnings for Dublin (€39,408) based on the Irish Central Statistics Office (CSO) Census figures [13].

**Table 1.** Characteristics of the sample.

|  | *n* | % | Variable | *n* | % |
|---|---|---|---|---|---|
| Gender |  |  | Marital Status |  |  |
| Male | 241 | 55.92 | Single | 208 | 48.26 |
| Female | 190 | 44.08 | Cohabiting | 75 | 17.40 |
| Total | 431 | 100.00 | Divorced | 13 | 3.02 |
| Age |  |  | Widowed | 2 | 0.46 |
| 18–24 | 162 | 37.59 | Domestic | 30 | 6.96 |
| 25–39 | 162 | 37.59 | Married | 103 | 23.90 |
| 40+ | 107 | 24.83 | Total | 431 | 100.00 |
| Total | 431 | 100.00 | Children/dependents (State) |  |  |
| Education |  |  | None | 295 | 68.45 |
| No formal education | 2 | 0.46 | 1 | 53 | 12.30 |
| Primary | 5 | 1.16 | 2 | 50 | 11.60 |
| Secondary | 115 | 26.68 | 3 | 19 | 4.41 |
| Technical | 30 | 6.96 | 4 or more | 14 | 3.25 |
| Advanced certificate | 26 | 6.03 | Total | 431 | 100.00 |
| Higher certificate | 52 | 12.06 | Income |  |  |
| Bachelors | 115 | 26.68 | 24,999 or less | 189 | 43.85 |
| Postgrad | 79 | 18.33 | 25,000–49,999 | 161 | 37.35 |
| Ph.D. | 7 | 1.62 | 50,000–74,999 | 60 | 13.92 |
| Total | 431 | 100.00 | 75,000–99,999 | 13 | 3.02 |
|  |  |  | 100,000 or more | 8 | 1.86 |
|  |  |  | Total | 431 | 100.00 |

The travel characteristics results (Table 2) showed that the largest share of the sample (31%) drove to work or education, followed jointly by pedestrians and bus, minibus, or coach (20%). Only 0.7% of the sample travelled by electric scooter and 0.5% by electric bike, which is to be expected given that at the time the survey was conducted, electric scooters were still illegal and not legislated for use on Irish roads; however, 8% of the sample were cyclists. The high share of private car users is compounded by more than half of the sample stating that they have free parking at their workplace or place of education (61%) and 81% of the sample having at least one private car available to their household. This determinant of mode choice is likely to be higher again for households with higher levels of car ownership. The results of this survey found that 44% of respondents reported having two or more cars available to the household.

**Table 2.** Travel characteristics of the sample.

| Variable | Survey | | Variable | Survey | |
|---|---|---|---|---|---|
|  | *n* | % |  | *n* | % |
| Mode |  |  | Car ownership |  |  |
| On foot | 84 | 20.34 | None | 83 | 19.26 |
| Bicycle | 32 | 7.75 | One | 157 | 36.43 |
| Electric bike | 2 | 0.48 | Two | 142 | 32.95 |
| Electric scooter | 3 | 0.73 | Three or more | 49 | 11.37 |
| Bus, minibus, or coach | 84 | 20.34 | Total | 431 | 100.00 |
| Train or tram | 53 | 12.83 |  |  |  |
| Motorcycle | 3 | 0.73 | Travel time |  |  |
| Driving a car | 129 | 31.23 | Less than 15 min | 64 | 15.50 |
| Electric car/alternatively fuelled car | 1 | 0.24 | 15–30 min | 193 | 46.73 |
| Passenger in a car | 11 | 2.66 | 30–45 min | 67 | 16.22 |
| Work from home | 10 | 2.42 | 45 min or more | 89 | 21.55 |
| Other | 1 | 0.24 | Total | 413 | 100.00 |
| Total | 413 | 100.00 |  |  |  |

**Table 2.** *Cont.*

| Variable | Survey | | Variable | Survey | |
|---|---|---|---|---|---|
| | *n* | % | | *n* | % |
| | | | Driver licence | | |
| Daily trip cost | | | Yes | 294 | 68.21 |
| €1 or less | 113 | 27.36 | No | 137 | 31.79 |
| €1–€5 | 184 | 44.55 | Total | 431 | 100.00 |
| €6–€10 | 93 | 22.52 | | | |
| €11–€15 | 14 | 3.39 | Free parking | | |
| €16+ | 9 | 2.18 | Yes | 262 | 60.79 |
| Total | 413 | 100.00 | No | 169 | 39.21 |
| | | | Total | 431 | 100.00 |

In order to generate an understanding of first and last mile access times to public transport services in the study area, which is often cited as a suitable micromobility trip condition [14,15], the respondents were asked to state how long (in minutes) it takes them to reach the nearest bus or train stop/station from their home, and from their workplace or place of education, which in both instances would be classified as being the first mile of a journey. As shown in Table 3, it was found that 88% of the sample have a 5 min or more walk access time to their nearest rail transport stop/station from their home, and 51% need to walk this same length of time to reach the nearest bus stop. Only 15% of the sample were 1–2 min from their nearest bus stop from their home. From their place of work or education, 57% were 5 min or more to the nearest rail stop/station, while 48% were 5 min or more to the nearest bus stop. However, 19% of respondents had a bus stop 1–2 min away vs. only 11% for rail. This highlights that a large proportion of commuting population may have a lengthy walk link in order to access existing public transport services, which is an ideal use case for micromobility, applying it to the pre-existing challenge for public transport operators to extend their coverage and patronage via enhancing accessibility.

**Table 3.** Access distances to nearest bus or rail station from home and place of work/education.

| From Home | Bus | Rail | From Work/Education | Bus | Rail |
|---|---|---|---|---|---|
| 1 min or less | 10% | 2% | 1 min or less | 5% | 6% |
| 1–2 min | 15% | 4% | 1–2 min | 19% | 11% |
| 2–3 min | 12% | 0% | 2–3 min | 11% | 13% |
| 3–4 min | 12% | 6% | 3–4 min | 17% | 13% |
| 5 min or more | 51% | 88% | 5 min or more | 48% | 57% |

This is compounded by the results for the question where respondents were asked to select which trip attribute they valued most in deciding what mode of transport to choose. The results of this question (see Table 4) showed that time was the primary influence on what mode of transport is chosen amongst the sample (32%), followed by convenience (27%), which was seen by the sample as being a larger influence than cost (19%) in terms of mode choice. This finding is consistent with other papers on the topic of electric scooters and micromobility, which determine that time savings or distance followed by convenience/enjoyment and then cost are the main motivations for mode choice, particularly for micromobility devices [6,7]. The weather and good infrastructure were viewed as being the least important factors in mode choice (4% and 1%, respectively). The sample were then also asked to state the distance and length of time of their (pre-COVID-19) commute to their place of work or education, which found that 213 respondents in the sample have a commute of 5 km or more, followed by 68 respondents stating that their commute distance is 2 km or less.

**Table 4.** Most valued trip attributes.

| Factors | *n* | % |
| --- | --- | --- |
| Time | 134 | 32.45 |
| Cost | 80 | 19.37 |
| Frequency | 21 | 5.08 |
| Comfort | 34 | 8.23 |
| Convenience | 111 | 26.88 |
| Good infrastructure | 4 | 0.97 |
| Weather | 15 | 3.63 |
| Other | 14 | 3.39 |
| Total | 413 | 100 |

In relation to electric scooters, the sample were asked how long they would walk in order to access or reach an electric scooter service. The results presented in Table 5 show that those in the 25–39 and 40+ age cohorts would be prepared to walk 5 min or more to such a service, which accounted for 40% and 49% of these age categories, while only 26% of 18–24-year-olds would be willing to walk this amount of time; this age group were more interested in a 2–3 min walk. More females were prepared to walk for longer (5 min or more) to access an e-scooter service than males (39% vs. 34% males). It was found that 4 min was the average amount of time that the sample would be willing to walk to access a shared scooter service.

**Table 5.** Cross tabulation of age and gender vs. access time to e-scooter service.

| Access Time | Age | | | Gender | |
| --- | --- | --- | --- | --- | --- |
| | 18–24 | 25–39 | 40+ | Female | Male |
| 1 min or less | 15% | 17% | 15% | 14% | 18% |
| 2–3 min | 19% | 5% | 8% | 12% | 10% |
| 3–4 min | 22% | 15% | 12% | 15% | 20% |
| 4–5 min | 19% | 23% | 16% | 20% | 19% |
| 5 min or more | 26% | 40% | 49% | 39% | 34% |

Respondents were similarly asked how much they would be willing to pay for a shared e-scooter service (per ride) in Dublin. Table 6 shows that 28% of females were willing to pay €1–2 for a shared e-scooter service and 10% were prepared to pay €4 or more, while 29% of males were willing to pay €1–2 and only 6% would pay €4 or more for this service. To determine the willingness to pay for an e-scooter service vs. the current daily trip cost of the sample, it was found that 49% of the sample who had a daily trip cost of €1–5 would be willing to pay €4 or more for a shared e-scooter service, while 45% would be willing to pay €3–4 per use. Of the respondents who have a daily trip cost of €1 or less, 31% stated that they would be prepared to pay €2–3 to use the shared e-scooter scheme.

**Table 6.** Willingness to pay vs. daily travel cost.

| Willingness to Pay | Gender | | Daily Travel Cost | Willingness to Pay for a Shared E-Scooter Service | | | | |
| --- | --- | --- | --- | --- | --- | --- | --- | --- |
| | Female | Male | | €1 or less | €1–2 | €2–3 | €3–4 | €4 or more |
| €1–2 | 28% | 29% | €1 or less | 37% | 27% | 31% | 15% | 11% |
| €1 or less | 23% | 28% | €1–5 | 45% | 52% | 34% | 45% | 49% |
| €2–3 | 22% | 24% | €6–10 | 13% | 19% | 29% | 30% | 31% |
| €3–4 | 16% | 12% | €11–15 | 2% | 2% | 5% | 5% | 3% |
| €4 or more | 10% | 6% | €16+ | 3% | 0% | 1% | 5% | 6% |

When asked to choose what the main benefits of electric scooters are, (see Table 7) 34% of sample stated that scooters are a useful alternative mobility option for people, followed

by 28% stating that they reduce their carbon footprint. Of the options available, the least popular benefit of electric scooters was that they are a fun mode of transport.

**Table 7.** Chosen benefits of electric scooters.

| Benefits of Electric Scooters | % |
|---|---|
| Electric scooters provide a useful alternative mobility option for people | 33.76 |
| Electric scooters reduce my carbon footprint | 27.77 |
| Electric scooters enhance accessibility to public transport stops/stations (i.e., first and last mile) | 21.15 |
| Electric scooters are fun, an aspect often missing in other transportation modes | 17.32 |
| Total | 100.00 |

Those in the sample who stated that they owned an e-scooter or had used an e-scooter sharing service were then asked what purposes they use their e-scooter for. The results showed that 27% of respondents used e-scooters for commuting to work, shopping/running errands, and socialising/visiting friends and family, while 15% used them as a commuting mode to education and 4% used them for other recreational activities. In these cases, 57% of respondents used the scooters several times a week, while 21% used them daily or less than once a month. If an electric scooter was not available to the respondents, it was found that 36% would have walked, 29% would have driven a car or cycled, and 7% would have taken public transport, which suggests that there is modal shift potential from private cars to e-scooters, notwithstanding the illegal status of e-scooters in Ireland when this study was conducted.

In relation to positive and negative attitudes towards e-scooters, as shown in Table 8, respondents reported that the benefits of e-scooters were that it is quicker than walking (22%), fun and relaxing (22%), and convenient to use (17%). Environmental reasons were chosen by 7% of sample, which was seen as of lesser importance than the convenience and time saving nature that micromobility presents. This supports the findings outlined earlier in the paper in relation to the main factors which influence mode choice, which were time savings and convenience. Safety concerns associated with riding e-scooters was cited as the main negative aspect reported by 34% of sample, which is a topic that is at the top of the agenda for micromobility providers and policy makers/adviser to address [16,17], followed by the charge/price per use (25%) and issues regarding the format of payments for e-scooters (18%).

**Table 8.** Positive and negative aspects of e-scooter experience.

| Positive Aspects of E-Scooter Experience | % | Negative Aspects | % |
|---|---|---|---|
| It was convenient | 17 | It was slow and inconvenient | 7 |
| It was fun/relaxing | 22 | It was expensive | 25 |
| It was quicker than walking | 22 | It was dangerous/unsafe | 34 |
| It was cheap/good value for money | 10 | It was complicated to pay for it | 18 |
| It was easier to get around than by driving | 7 | It was difficult to ride | 9 |
| It was easier to get around than by PT | 7 | Other | 7 |
| It was comfortable/easy to ride | 6 | Total | 100 |
| It is good for the environment | 7 | | |
| Other | 2 | | |
| Total | 100 | | |

A binary logit model was conducted to examine the statistical influence of various independent variables (socio-demographic and trip variables) on mode choice. The results from this analysis, presented in Table 9, show that longer trip distances, higher trip costs, and car ownership influenced the choice of private car. Other characteristics which were linked to car mode choice were middle aged females with at least one child.

For bus mode choice, the results showed that males with no children/dependents with short to medium-distance trips were linked to choosing public transport. As expected,

car ownership and higher incomes negatively impacted the likelihood of this choice taking place.

Men were more likely to walk and/or cycle to work than females, and as anticipated, longer-distance trips, having a car available, higher income, and having children/dependents negatively influenced the walking mode choice. In fact, medium-distance trips, having an average annual income and level of education, and having a driver licence were determinants of choosing to cycle to work.

As expected, younger age cohorts were more likely to choose an e-scooter, and those in higher income brackets were also more likely to use an e-scooter. Furthermore, short trip times, not owning a driver licence or a car, and those with higher trip costs were associated with e-scooter mode choice. The statistical significance of many of the e-scooter coefficients is poor due to the low number of respondents that chose e-scooter for commuting trips and, of course, the illegal status of e-scooters in Ireland at the time of writing.

The final section presented in this paper outlines a wider discussion of the results presented from the study in addition to how such results compare or contrast to studies from other countries and recommendations for future research on this topic.

**Table 9.** Binary logit model results.

| Variable | Private Car | | Public Transport | | Cycling | | Walking | | E-Scooter | |
|---|---|---|---|---|---|---|---|---|---|---|
| | **B** | ***p*-Value** | **B** | ***p*-Value** | **B** | ***p*-Value** | **B** | ***p*-Value** | **B** | ***p*-Value** |
| Constant | −4.076 | 0.000 | −4.342 | 0.000 | −0.914 | 0.454 | 3.134 | 0.002 | −3.918 | 0.281 |
| Age | 0.841 | 0.001 | −0.567 | 0.015 | −0.306 | 0.472 | −0.415 | 0.264 | −3.242 | 0.085 |
| Gender | −0.304 | 0.064 | 0.111 | 0.411 | 0.082 | 0.716 | 0.063 | 0.751 | 0.125 | 0.852 |
| Education | −0.107 | 0.206 | 0.047 | 0.507 | 0.275 | 0.031 | 0.068 | 0.507 | 0.192 | 0.649 |
| Children | 0.132 | 0.430 | 0.040 | 0.802 | 0.381 | 0.153 | −0.261 | 0.310 | 0.719 | 0.328 |
| Marital status | −0.036 | 0.694 | −0.086 | 0.286 | −0.330 | 0.045 | 0.116 | 0.354 | 0.136 | 0.746 |
| Driving licence | −3.070 | 0.000 | 0.700 | 0.028 | 1.048 | 0.037 | 0.182 | 0.683 | −0.343 | 0.839 |
| Free parking | −1.207 | 0.001 | 1.160 | 0.000 | 0.493 | 0.314 | −0.115 | 0.790 | 0.874 | 0.549 |
| Car ownership | 0.664 | 0.000 | −0.314 | 0.014 | −0.234 | 0.277 | −0.305 | 0.138 | −0.55 | 0.475 |
| Distance | 0.684 | 0.000 | 0.542 | 0.000 | 0.798 | 0.000 | −1.929 | 0.000 | 0.316 | 0.601 |
| Trip time | −0.922 | 0.000 | 0.340 | 0.007 | −0.590 | 0.016 | 1.510 | 0.000 | −1.206 | 0.123 |
| Income | 0.062 | 0.734 | −0.106 | 0.548 | 0.130 | 0.629 | −0.164 | 0.459 | 1.239 | 0.053 |
| Trip cost | 0.878 | 0.000 | 0.565 | 0.001 | −2.419 | 0.000 | −1.401 | 0.000 | 0.891 | 0.264 |
| Log Likelihood | 272.233 | | 369.383 | | 150.626 | | 180.244 | | 25.862 | |
| Nagelkerke R Square | 0.618 | | 0.435 | | 0.392 | | 0.686 | | 0.280 | |

## 4. Discussion and Conclusions

This study has provided some insights into the existing and potential future demand of electric scooters in Dublin and the attitudes and perceptions of Dubliners towards e-scooters. The main takeaway points produced from this study include the findings that more than a quarter (27%) of e-scooter users in the sample used them as a commute mode for travelling to and from work, shopping, or running errands, whereas 15% used them for travelling to and from college/university. In relation to frequency of use, over half (57%) of e-scooter riders used them several times in a week, and if an e-scooter was not available, 29% of the users would have driven a private car, 29% would have cycled, 36% would have walked, and 7% would have taken public transport. Time followed closely by convenience were the two attributes that influenced the choice of transport mode most, which suggests that people value their time more than the cost associated with the mode. Meanwhile, younger people, those with higher incomes, and those without a driver licence or private car were more likely to opt for an e-scooter, and shorter trip time was associated with e-scooter mode choice in the binary logit model. The main benefits of e-scooters reported by respondents in the study were that they are quicker than walking, fun and relaxing, and convenient to use. Finally, those in the sample who have a daily trip cost of €1–5 were found to be willing to pay €4 or more for a shared e-scooter service, and 31%

respondents with a travel cost of €1 or less would be prepared to pay €2–3 for the scheme, which suggests that people would be willing to increase their daily travel costs in order to use the shared e-scooter service.

### 4.1. Summary and Conclusions

Ultimately, there is an inherent need to examine the situation of e-scooter use as a whole more broadly and encourage open-mindedness in first understanding where novel modes such as e-scooters and other micromobility devices could fit into the mobility fabric (or not) of our towns and cities, recognising that while e-scooters present certain challenges from a regulatory and safety standpoint, such challenges could be surmountable via clear, enforced regulation and laws and a dialogue between micromobility and public transport operators, local authorities, and county/city councils. Given that 29% of e-scooter users in the sample would have driven otherwise, it is indicated that if implemented appropriately, e-scooters could offer an attractive alternative to private car use for certain trip purposes, which is compounded by the largely positive attitudes of the sample towards e-scooters. This suggests that there is some potential to encourage modal shift that can reduce car dependency and ease traffic congestion. This could be enhanced by raising awareness and education (particularly at secondary school levels) of electric micromobility and the safe and responsible use of e-scooters among the public when legalised, as suggested by Turoń et al. [18] in Poland. This can also be achieved by simply enhancing accessibility to existing public transport services for first and last mile trips or replacing the car outright for short to medium-length trips, which were found to be 2 km or less (10 min or less) for over half of the sample (54%) and 3–4 km for 43% of the sample (11–20 min). This corresponds to findings from Ayfantopoulou et al. [19] in Thessaloniki, Greece, where a distance of 1–6 km or travel times of less than 15 min were found to be most suitable in terms of convenience. Conversely, given that 36% of respondents stated that if an e-scooter was not available, they would walk, and 29% would cycle, it is also pertinent to acknowledge that some studies have highlighted the negative environmental impact of replacing walking and non-electric bicycle trips with e-scooters, such as in North Carolina, USA [20], which was found to be higher for shared users than private users. The difference between the potential for shared vs. private e-scooters replacing private cars is a topic that needs further investigation. However, this is also dependent on trip purpose, as Moreau et al. [21] found that personal e-scooters are used more frequently for commute trips than leisure trips, which are more popular with shared services.

Secure parking/storage for private and public (shared) bikes, e-bikes, and e-scooters at bus and rail stations at both ends of the public transport trip to cater for the first and last mile segment of the door-to-door journey, particularly in neighbourhoods with disproportionately high percentages of short to medium-length private car trips, could see the greatest benefit from micromobility deployments in terms of reducing car dependency, alleviating congestion, and reducing tailpipe emissions from private cars. When integrated with public transport (PT) modes, micromobility can provide seamless door-to-door services as an alternative to a private car, which presents a sustainable and equitable means of encouraging a modal shift by extending the catchment area of existing bus and rail services, thereby enhancing accessibility. In this way, micromobility may assist in reducing the transport access barrier that exists in 'transit deserts' [21].

### 4.2. Plans for Future Research and Limitations of the Study

It is acknowledged that further research is required to generate a richer understanding and representation of micromobility user behaviour using operator/service provider data when e-scooters do become legalised in the Republic of Ireland and e-scooter providers begin operations in Dublin, amongst other Irish cities. For future research, it is recommended a study be conducted to compare the attitudes and perceptions of the people of Ireland before and after the legislation is passed to regulate e-scooters and permit their use on public roads. The study could also be extended to other cities/regions of Ireland without

a focus on County Dublin, which was a limitation of the study given funding and time restraints regarding collecting a larger sample size that would be nationally representative. It is also acknowledged that micromobility devices, particularly e-scooters, are not without their issues, often in relation to age limits, location and specification of parking facilities (i.e., station-less/virtual vs. stationed), vehicle collisions, and near misses with pedestrians and other road users. In the case of Ireland, akin to the situation in the UK, these issues have stifled advancement toward legalization of the devices on public roads. However, when forthcoming legislation is presented, it will likely clearly define the legal use case of micromobility devices and outline how concerns, often raised by vulnerable road users, will be managed, such as banning use on footpaths, regulating speed, age limitations, and commonplace rules on having functioning lights on the devices in the dark. The reflections of the concerns outlined in this paper as well as the potential benefits that micromobility can present to the sustainable mobility offering, both for private and shared users, will be of interest to other policymakers, researchers, and local government officials in devising similar regulations and laws surrounding e-scooter use in cities in Europe and other regions globally. Time will tell whether this 'final frontier' of legalised e-scooter use in Europe will be reached or not; however, ultimately, a wider array of alternatives to the private car being available to travellers for all trip purposes arguably presents a valid opportunity to reduce sustained car use for short to medium-distance trips.

**Funding:** This research was funded by [Zipp Mobility] grant number [0123].

**Informed Consent Statement:** Not applicable.

**Data Availability Statement:** Data available on request due to restrictions eg privacy or ethical.

**Conflicts of Interest:** The author declares no conflict of interest. The funders had no role in the design of the study; in the collection, analyses, or interpretation of data; in the writing of the manuscript; or in the decision to publish the results.

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
