# Peer review of "Perceptions of Electric Scooters Prior to Legalisation: A Case Study of Dublin, Ireland, the ‘Final Frontier’ of Adopted E-Scooter Use in Europe"

_sustainability, doi:10.3390/su141811376_

Round 1

Reviewer 1 Report

The article concerns an interesting topic which is "Perceptions of electric scooters prior to legalisation, a case study of Dublin, Ireland, the ‘final frontier’ of adopted e-scooter use in Europe". The article is interesting for the reader and it refers to up-to date topic of new mobility solution which is scooter-sharing.

The article deserves to be published and fully fits into the thematic scope of the journal and the special issue.

The language is correct. The literature review is correctly described and shows a large number of references. The methodology is described succinctly and correctly. 

I have a few considerations to improve the quality of the article.

1. In the introduction, please also critically address the scooter sharing services and the problems with the proper functioning of the systems occurring there. For interesting research on this subject, see 'The Concept of Rules and Recommendations for Riding Shared and Private E-Scooters in the Road Network in the Light of Global Problems'.

2. In the discussion, please refer to the research carried out by other scientists on the same subject and confirm or deny the similarity of the results. It would also be interesting to discuss the results in relation to other countries.

3. In the chapter with recommendations, indicate specific recommendations from the conducted research. 

4. Please add a summary chapter. 

5. In the conculsions in line 455 "This suggests that there is some potential to encourage modal shift potential that can reduce car dependency and ease traffic congestion. This could be enhanced by addressing added incentives provided to private car users such as free workplace car parking or free parking available at places of education/study, which was found to be one of the main determinants of private car mode choice for commute trips in this paper." please refer to the need to raise awareness of electric mobility and the use of e-scooters among the public. Check ' When, What and How to Teach about Electric Mobility? An Innovative Teaching Concept for All Stages of Education: Lessons from Poland '.

6. In summary, also point to future research. Do you plan to expand them, what should be paid attention to in this matter.

7. In summary also add information about the limitations of the study.

Good luck!

Reviewer 2 Report

The topic of the article is topical and relevant, but it is focused only on a very small area. The article presents the results of studies carried out during the epidemic, the traffic volume in this period was definitely lower, so it is worth referring these results to the time "after the pandemic". The paper presents an extensive analysis of the literature, but it is difficult to relate it to the research conducted by the authors. As mentioned, the work only refers to the Dublin area, which means that the manuscript has no global significance and cannot be applied to other places. In addition, the work identifies local micromobility laws, which are different for each region. The work includes questionnaire research and their basic statistical analysis. The presentation can be of great value in practice, but only locally. The manuscript does not add value to the discipline.

Reviewer 3 Report

comments as attached 

Round 2

Reviewer 1 Report

The article has been significantly improved and in my opinion is suitable for publication. 

My sincere congratulations on such a valuable article that fills the research gap in the field of e-scooters. 

Author Response

Thank you very much for taking the time to review my manuscript and for your comments that have improved the quality of the paper.

Reviewer 2 Report

The corrections made are sufficient. However, 

Author Response

(The authors gave the same response as above.)

Reviewer 3 Report

see attachment

Round 3

Reviewer 3 Report

comments as attached

Author Response

Thank you again for reviewing the manuscript.

The qualitative opinions and attitudes section of the paper has now been removed from the manuscript.